

# The PolarRES dataset: a state-of-the-art regional climate model ensemble for understanding Antarctic climate

**Ella Gilbert** ♦[1]**, José Abraham Torres-Alavez**♦[2]**; Marte G. Hofsteenge**♦*[3]**;** Willem Jan van de Berg[3]; Fredrik Boberg[2]; Ole Bøssing Christensen[2]; Christiaan Timo van Dalum[3,4]; Xavier Fettweis[5]; Siddharth Gumber[1]; Nicolaj Hansen[2]; Christoph Kittel[6,7]; Clara Lambin[5]; Damien Maure[5]; Ruth Mottram[2]; Martin Olesen[2]; Andrew Orr[1]; Tony Phillips[2]; Maurice van Tiggelen[3]; Kristiina Verro[2] & Priscilla A. Mooney[8]

♦ These authors contributed equally to leading this work

* *Correspondence to:* Ella Gilbert (ellgil82@bas.ac.uk) and Marte Hofsteenge (m.g.hofsteenge@uu.nl)

[1] British Antarctic Survey, Cambridge, United Kingdom
[2] Danish Meteorological Institute, Copenhagen, Denmark
[3] Institute for Marine and Atmospheric Research (IMAU), Utrecht University, Utrecht, the Netherlands
[4] Royal Netherlands Meteorological Institute, De Bilt, the Netherlands
[5] SPHERE research unit, Department of Geography, University of Liège, Liège, Belgium
[6] Université Grenoble Alpes/CNRS/IRD/G-INP/INRAE, Institut des Geosciences de l'Environnement, Grenoble, France
[7] Physical Geography research group, Department of Geography, Vrije Universiteit Brussel, Brussels, Belgium
[8] NORCE Norwegian Research Centre, Bjerknes Centre for Climate Research, Bergen, Norway

**Abstract.**

Antarctica's weather and climate have global impacts, influencing weather patterns, ocean currents and sea levels worldwide. However, Antarctica is vast and complex, and the atmospheric processes that govern its climate are strongly influenced by its steep terrain, particularly around the coastal periphery. Our scientific understanding of this complex environment is hampered by the lack of reliable observations and gridded datasets at sufficiently high spatial and temporal resolution. High-resolution regional climate models, RCMs, can provide a solution to the sparsity of observational data and low resolution of reanalyses, facilitating more in-depth assessments of crucial climate variables like precipitation, winds and temperatures that are strongly influenced by topography. Here we present and evaluate a comprehensive, high-quality, ~11 km resolution RCM dataset: the PolarRES ensemble. We show that the ensemble largely out-performs ERA5, particularly for variables such as coastal winds and in characterising high-resolution regional precipitation patterns. There are no consistent seasonal differences in biases, but there are persistent regional biases. The Victoria Land region is the region the RCMs and ERA5 struggle the most with, which suggests that further investigation and model development is needed in this area. Each RCM has strengths and limitations, but overall the ensemble captures the observed weather and climate of Antarctica well. The PolarRES ensemble offers a novel and exciting way of evaluating climate processes and features, and we encourage researchers to use the data, which are freely available, to explore pertinent climate questions of local, regional and global significance.



### Intro

Antarctica plays a critical role in the global climate system, impacting planetary energy balance, sea level rise, weather patterns, ocean and atmospheric circulation and ecosystems of global significance. However, although it is vitally important to understand the present-day climate of Antarctica and how it will change in future, both are uncertain, primarily because of the dearth of observations on this extreme continent. This lack of observations makes it especially challenging to understand temporal and spatial climate variability resulting from complex

atmospheric processes that operate at fine scales. Antarctica's weather and climate are strongly influenced by the complex topography, particularly around the periphery of the ice sheet, where the majority of precipitation and ice melting occurs (Nicola et al., 2023; Mottram et al., 2021). Orographic precipitation is a significant contributor to the continent's surface mass balance, and orography also generates localised atmospheric phenomena like katabatic winds and foehn winds. Both wind types are important features of the climate in coastal areas with steep

terrain and can influence the seasonal and annual climates of these regions. Foehn winds have been shown to impact the localised climate across Antarctica, for example over the Antarctic Peninsula (Datta et al., 2019; Gilbert et al., 2022a; 2022b; Wille et al., 2022), Amundsen Sea Embayment (Francis et al., 2023; Gilbert et al., 2025), Ross Ice Shelf (Hansen et al., 2024), McMurdo Dry Valleys (Speirs et al., 2010; Hofsteenge et al., 2022) and East Antarctica (Wille et al., 2024). Meanwhile, katabatic winds exert a dominant influence on the climate of coastal

Antarctica (Heinemann et al., 2019; 2021; Parish & Bromwich, 2003; Caton Harrison et al., 2024), including in regions such as Coats Land (Renfrew & Anderson, 2003) and the Eastern Antarctic peninsula, where they interact with barrier winds (van Wessem et al., 2015; Gilbert et al., 2022b). Representing the continent's complex orography is therefore crucial for accurately simulating these important features of the Antarctic climate.

Because of Antarctica's harsh environment and vast size it is difficult to collect in situ data. Consequently, the observational network is relatively sparse in both space and time, and so products like satellite data and atmospheric reanalysis are typically used to evaluate the spatial variability of important climate variables like temperature, humidity, winds, pressure and precipitation. However, satellite products are difficult to apply comprehensively because of issues like their lack of coverage at high latitudes, interference from cloud cover,

their difficulty in distinguishing between ice surfaces and cloud, and the assumptions made in the models used to translate the raw signals received into meaningful physical quantities (Gabarró et al., 2023). Although reanalyses can reliably capture large-scale features of the Antarctic climate, they have known biases. The three commonly used reanalyses in the Antarctic, ERA5, JRA-55 and MERRA2, all have different strengths and weaknesses. Caton Harrison et al. (2022) shows that of the three, ERA5 performs best at capturing low level coastal winds, partly

because it has the most realistic representation of orography, a finding also reported by Dong et al. (2020). ERA5 is also shown to capture seasonal variation in near-surface air temperature and accumulation well, while MERRA2 performs better with respect to winds in the Antarctic interior (Gossart et al., 2019). ERA5 is shown by Wang et al. (2025) to best represent surface mass balance and 10 m air temperature. Generally, reanalyses are at insufficient horizontal resolution to explore processes that are strongly dependent on local features. It is known that the

representation of orography can impact important climate features like the quantity and properties of simulated precipitation (Datta et al., 2023; Gilbert et al., 2025), and the important localised features noted above. Increasing model resolution is therefore a promising way to improve understanding of the Antarctic climate, particularly in these critical coastal regions.



Regional Climate Models (RCMs) can offer a substantial improvement in the representation of important processes relative to reanalysis products, especially at the sub-continental scale. RCMs offer both enhanced resolution and more sophisticated physical parameterisations, which can be tailored to the polar environment for greater accuracy. Many RCMs include additional processes that are vital for adequately representing the polar climate but which are missing from global reanalyses or global models. Examples include blowing and drifting

snow, snowpack evolution, surface snowpack hydrology (including refreezing, melt retention and runoff), sea ice processes, albedo and snow-on-sea ice modules. RCMs have also been shown to better capture extreme events, including extreme precipitation, extreme heat, atmospheric rivers, winds and melt events (Campbell et al., 2024; Iles et al., 2020; Gilbert et al., 2025; Kolbe et al., 2025). For several years, the polar regional climate modelling community has produced RCM simulations of the Antarctic climate as part of Polar CORDEX, a polar-specific

initiative of the Coordinated Regional Climate Downscaling Experiment. Mottram et al. (2021) and Hansen et al. (2022) demonstrated the value of the previous generation of Polar CORDEX RCMs – an ensemble of RCMs at 25-50 km resolution forced with ERA-Interim reanalysis (Dee et al., 2011) - for evaluating Antarctic surface mass balance. The latest generation of Polar CORDEX RCMs have been run at approximately 11 km resolution as part of the PolarRES project, an EU Horizon 2020 funded project that uses RCMs to simulate the current and future

climate of the polar regions. This jump in resolution represents a significant advance in our ability to simulate the climate of the Antarctic over long time periods, and therefore to evaluate present and future changes in the region.

This study presents an evaluation of these latest RCMs – the PolarRES ensemble – with respect to the near-surface climate of the recent past. Some evaluation of this ensemble has already been done on shorter timescales and over

specific regions, for example Gilbert et al. (2025) showed that during two case studies, a 'mini-ensemble' of three PolarRES RCMs at 12km resolution represented a substantial improvement relative to ERA5 when simulating total precipitation over the Amundsen Sea Embayment. Verro et al. (2025) evaluate the performance of a number of the PolarRES RCMs with respect to Antarctic sea ice albedo in two case studies. Price et al. (2024) assess the ability of the MetUM with various technical configurations to simulate cloud and aerosol processes over coastal

Antarctica, and van Dalum et al. (2025) evaluate the performance of RACMO2.4, the version used in this study, with respect to the near-surface climate and surface mass balance of Antarctica and Greenland. However, no study has yet evaluated these new models over the entire pan-Antarctic domain on the multi-decadal timescale. In this study we address this gap by assessing the performance of the PolarRES ensemble in the representation of near-surface meteorological variables. We compare against available weather station data and ERA5 with respect to

near-surface humidity, wind speed, air temperature, surface pressure and precipitation.

## 2 Methods and materials

### 2.1 Forcing data

ERA5 is the fifth-generation reanalysis produced by the European Centre for Medium-range Weather Forecasts, ECMWF, providing comprehensive global climate and weather data spanning the past eight decades, with records

available from 1940 onward. It serves as the successor to the ERA-Interim reanalysis. The dataset features a horizontal resolution of 0.25 degrees, which corresponds to approximately 31 kilometres, and includes 137 vertical levels. ERA5 is produced using 4D-Var data assimilation and model forecasts in cycle 41r2 of the ECMWF Integrated Forecast System (IFS) (Hersbach et al., 2020).



**2.2 The PolarRES ensemble**

The Antarctic ensemble comprises four RCMs: HCLIM, MAR, MetUM and RACMO, all of which are run for the recent past. All models have been used to simulate the period 2000-2019 inclusive, and several models have simulations extending further backwards and forwards in time. For this evaluation, we focus on the 20-year period 1 January 2000 – 31 December 2019. The RCMs are all forced with ERA5 reanalysis, described above, and nudged or re-initialised to ERA5 once per day throughout the simulation.


All four RCMs are run over the same region covering the Antarctic continent and Southern Ocean, as well as the very southern tip of South America: HCLIM, MetUM and RACMO use exactly the same domain, at a resolution of 11 km, while MAR uses a slightly smaller domain and a resolution of 12 km. Technical details of each RCM are given below; fuller details of the domains, parameterisation schemes and overall model experimental protocol

will be described in a forthcoming manuscript (PolarRES consortium, in prep.).

**2.2.1 HCLIM**

The regional atmospheric model HARMONIE-Climate (HCLIM; Belušić et al., 2020) is the climate configuration of the HARMONIE limited area numerical weather prediction system (Bengtsson et al., 2017). Surface processes are represented through the SURFEX (Masson et al. 2013) surface modelling system, which includes the 12-layer

Explicit Snow scheme of the ISBA land surface model (ISBA-ES; Boone and Etchevers, 2001; Decharme et al., 2016). Sea ice processes are represented using the one-dimensional thermodynamic Simple Sea-Ice Scheme (SICE; Batrak, 2021). For PolarRES, HCLIM is run with a horizontal resolution of 11 km and 65 vertical levels, using version cycle 43 with the hydrostatic ALADIN atmospheric physics. The model is forced with ERA5 reanalysis every three hours and uses spectral nudging of vorticity (e-folding time: 6 hours), air temperature (24

hours), and divergence (48 hours).

**2.2.2 MAR**

The Modèle Atmosphérique Régional (MAR) is a hydrostatic, primitive equation regional atmospheric model specifically developed for the polar areas (Gallée and Schayes, 1994). It is coupled with the SISVAT snow and ice module (De Ridder and Gallée, 1998) and includes a cloud microphysics scheme for conservation equations

for cloud droplets, rain drops, cloud ice crystals, and snow particles (Gallée, 1995). The new 3.13 version also includes the ecRad radiative scheme, described in Grailet et al. (2025). The PolarRES MAR domain is a polar stereographic 597x 531 grid at 12.5km resolution, with 24 vertical levels. The top 9 levels are nudged with ERA5 winds and temperature every 6 hours.

**2.2.3 MetUM**

The UK Met Office Unified Model (MetUM; Walters et al., 2019), is run in its regional configuration in atmosphere-only mode. It is a non-hydrostatic model with a fully compressible atmosphere, semi-Lagrangian advection and semi-implicit time-stepping, based on the ENDGame dynamical core. It is run in the same manner as in Gilbert et al. (2022a), with frequent re-initialisations from the forcing data in a pseudo-nudging approach (true nudging is unavailable in this version of the MetUM). The global model (at ~25 km resolution) is initialised

every 24 h from ERA5 at 12:00 UTC and this data is downscaled to the Antarctic limited area domain at 11 km. The limited area domain is used to run 36 h forecasts, of which only the t+12 to t+36 h forecasts are retained,





resulting in a continuous time series. The initial 12 h period is discarded as model spin up time. The GA7 science
configuration of the MetUM is used (Walters et al., 2019), which includes the PC2 large-scale cloud and saturation
scheme (Wilson et al., 2008a, 2008b) that simulates prognostic cloud fraction and prognostic condensate terms,

and a microphysics scheme based on the original single-moment scheme of Wilson and Ballard (1999), with
significant adaptations. A simple four-layer configuration of the JULES land surface scheme (Best et al., 2011;
Clark et al., 2011) is used, as limited work has been done to validate the scheme in the Antarctic and its complexity
consumes additional computational resources.

### 2.2.4 RACMO

The Regional Atmospheric Climate MOdel (RACMO), version 2.4p1 (van Dalum et al., 2024) is a limited area
model combining the semi-Lagrangian HIRLAM dynamical core (Undén et al., 2002) with the physical
parameterisations for sub-grid scale processes of cycle47r1 of the IFS model (ECMWF, 2020). IFS physical
parameterisations include processes like convection, turbulence and surface processes and the radiation scheme
ecRad (Hogan and Bozzo, 2016). RACMO uses semi-implicit time-stepping and a hydrostatic approach. At the

lateral boundaries, RACMO is forced by ERA5 for wind speed, pressure, humidity and temperature, and at the
sea-surface interface for sea surface temperature and sea-ice extent. In this RACMO version, 40 atmospheric
layers are used and the upper layers are nudged (van de Berg & Medley, 2016).

The polar version of RACMO is optimised specifically for the polar regions and includes a dedicated surface tile

for glaciated areas. Parameterisations have been developed for this tile, such as a multi-layer snow module, a
spectral snow albedo and radiative transfer scheme, snow densification and metamorphism, and a melt water
scheme. For full details, please refer to van Dalum et al. (2024). For experiments in this study, we use RACMO
model output produced by van Dalum et al. (2025) on an 11 km horizontal-resolution grid.

### 2.3 Evaluation datasets

The AntAWS dataset (Wang et al., 2022) is the primary observational dataset used to evaluate the PolarRES
ensemble. The dataset collates 267 in situ automatic weather stations from across the Antarctic continent, and
contains measurements of air temperature, surface pressure, relative humidity, wind speed and wind direction at
3-hourly, daily and monthly intervals, plus quality control information. We use daily data for comparison with the
daily model outputs.


We use a subset of the stations contained within the dataset, because we chose to use only stations with more than
60% temporal coverage. That is, only stations which have valid data for > 60% of the time period of interest. This
allows us to maintain a sufficient number of stations around the continent while ensuring the seasonal mean of
the variables is statistically significant. The stations that are included in the analysis are shown as red dots on

Figure 1.

To account for differences in surface elevation between the RCM and observations at the location of the AWSs,
we apply a correction. Modelled near-surface temperature and surface pressure are corrected for elevation
differences using the dry-adiabatic lapse rate and assuming a standard atmosphere.




We also compare the RCM outputs directly to ERA5, but it must be stressed that this is not an independent comparison because the RCMs are all forced by ERA5. This analysis therefore tells us only about the differences introduced by the RCM model physics and improved resolution compared to the driving data.

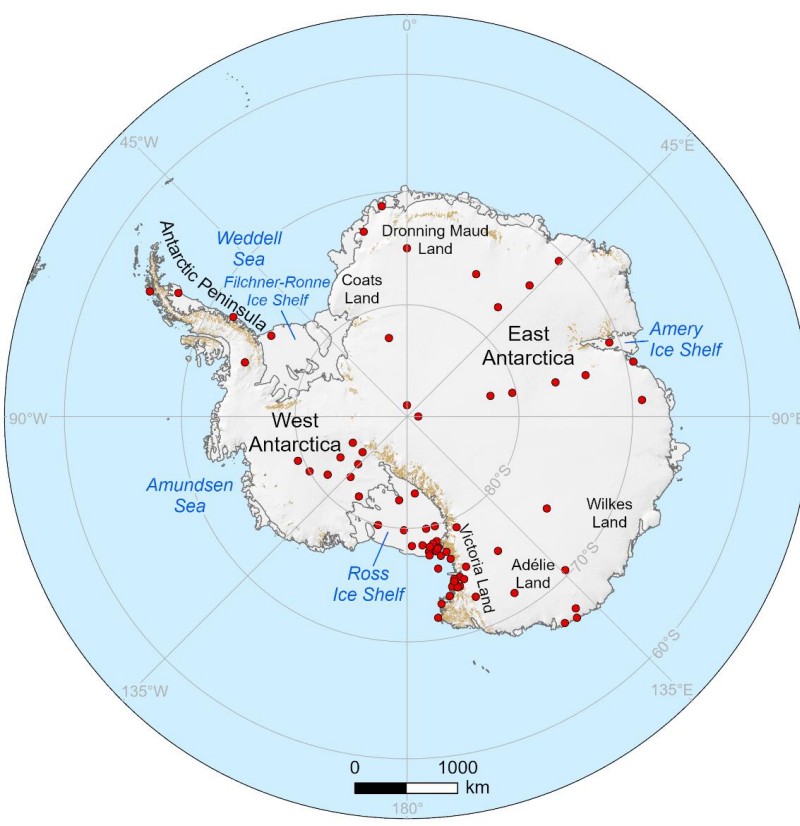


**Figure 1: Map of the Antarctic region, including the stations used in the analysis. Underlying data from the SCAR Antarctic Digital Database, 2024.**





**Figure 2: Multi-model mean (MMM) and standard deviation (SD) of the RCM ensemble for the summer (DJF) and winter (JJA) seasons. We show, from top to bottom, near-surface temperature, near-surface specific humidity, 10 m wind speed, surface pressure and surface precipitation.**





**Table 1: Mean near-surface climate over the Antarctic land area as simulated by ERA5, the four regional climate models (RCMs), and their multi-model mean (MMM). For each model and the MMM, averages are shown over the Antarctic land area for austral summer (DJF) and winter (JJA); for precipitation, continent-integrated totals are shown (Gt).**

|  | ERA5 | | HCLIM | | MAR | | MetUM | | RACMO | | MMM | |
|---|---|---|---|---|---|---|---|---|---|---|---|---|
|  | *DJF* | *JJA* | *DJF* | *JJA* | *DJF* | *JJA* | *DJF* | *JJA* | *DJF* | *JJA* | *DJF* | *JJA* |
| **Temperature [°C]** | -23.3 | -42.5 | -15.6 | -41.6 | -21.8 | -40.3 | -21.4 | -41.7 | -23.0 | -43.0 | -20.4 | -41.7 |
| **Specific Humidity [g kg⁻¹]** | 0.73 | 0.15 | 1.43 | 0.19 | 0.87 | 0.21 | 1.00 | 0.24 | 0.79 | 0.18 | 1.02 | 0.21 |
| **Wind Speed [m s⁻¹]** | 5.8 | 8.0 | 4.2 | 7.0 | 6.0 | 8.7 | 6.0 | 8.7 | 5.6 | 7.9 | 5.4 | 8.1 |
| **Pressure [Pa]** | 762 | 754 | 777 | 772 | 779 | 774 | 780 | 773 | 778 | 772 | 779 | 772 |
| **Precipitation [Gt]** | 395 | 529 | 776 | 684 | 504 | 724 | 600 | 724 | 573 | 758 | 613 | 772 |


### 3. Results

#### 3.1 Mean climate and ensemble spread of RCMs

Figure 2 and Table 1 summarise the mean meteorology of the ensemble. Figure 2 shows the RCM multi-model mean (MMM) and standard deviation of five important near-surface variables in the ensemble during summer and
winter. Meanwhile, Table 1 shows the Antarctic-averaged values of the variables considered for ERA5 and each RCM.

Table 1 shows that MMM summer (DJF) temperatures over the Antarctic continent are –20.4 °C (-23.0°C to –15.6°C) while MMM winter (JJA) temperatures are –41.7°C (-43.0°C to –40.3°C). However, as shown in the top
row of Figure 2, the magnitude of this seasonal temperature difference varies spatially. MMM temperatures vary between DJF and JJA by approximately 40°C over the Antarctic Plateau as radiative cooling during the polar night causes the surface to lose heat. Meanwhile, seasonal temperature differences are comparatively smaller near the coasts. Temperatures over the open ocean are much warmer in both seasons, with cooler temperatures over the sea ice area due to radiative cooling from the ice surface. Along with the comparative summer warmth, the
RCMs also simulate much higher MMM summertime specific humidity (1.02 g kg⁻¹ compared with 0.21 g kg⁻¹ in winter when averaged across the whole continent: Table 1), as the warmer atmosphere has a higher specific heat capacity, as shown in the second row of Figure 2 and Table 1. MMM wind speeds (third row) are strongest around the coasts - especially in the Eastern sector of the continent - during both seasons, with winds generally higher during JJA, reaching maximum MMM values of up to around 10 m s⁻¹ in DJF and 20 m s⁻¹ in JJA. MMM values,
as averaged across the continental land area, are 5.4 m s⁻¹ (4.2 m s⁻¹ to 6.0 m s⁻¹) in DJF compared with 8.1 m s⁻¹ (7.0 m s⁻¹ to 8.7 m s⁻¹) in JJA (Table 1). Surface pressure, shown in the fourth row of Figure 2, varies comparatively little between summer and winter, as seasonal differences are relatively small compared to the spatial variability



caused by topography (Table 1). MMM precipitation is highest on the Western side of the Antarctic Peninsula and along the coast of West Antarctica, and is higher in JJA than DJF, reaching seasonal mean values over 1000

mm w.e., or 772 Gt over the season (Table 2).

Figure 2 also shows standard deviations for each of these variables, showing the degree of variability between RCMs. The standard deviation of the ensemble's temperature is largest during DJF over the Ross, Filchner-Ronne and Amery ice shelves, as well as the Victoria Land region on the eastern side of the Ross Ice Shelf and Weddell

Sea. Standard deviations reach maximum values of around 6°C over the large ice shelves. The Victoria Land region stands out even more during JJA, with standard deviations of up to 10°C, likely due to differences in how the RCMs simulate atmospheric interactions with complex topography here and resulting adiabatic processes. Much of the standard deviation in temperature – particularly in DJF – comes from the higher temperatures at low elevation in HCLIM (not shown; see below for further discussion). In DJF, ice shelves and other low-elevation

areas around the continent's periphery appear with the largest standard deviations in relative humidity, and sea ice areas in the Weddell Sea also stand out due to the humidity values simulated by MAR deviating from the ensemble mean (not shown), particularly the Ross, Filchner-Ronne and ice shelves abutting the Weddell Sea. Maximum DJF standard deviation values for relative humidity exceed 0.6 g kg$^{-1}$ over the Filchner-Ronne and Ross Ice Shelves, however, they are universally below 0.1 g kg$^{-1}$ across most of the domain in JJA. Standard

deviations in wind speed are higher in winter when wind speeds are higher, reaching 5 m s$^{-1}$ over Victoria Land coasts in JJA, with the eastern Antarctic Peninsula, a region with a dominant wintertime barrier wind regime (Gilbert et al., 2022b) also standing out as a region of higher intra-model variance. In DJF, wind speed standard deviations are higher along the coast of East Antarctica from Dronning Maud Land to Adelie Land – again most of this variance comes from HCLIM (not shown). For surface pressure, standard deviations are more spatially

mixed in both seasons, while there are seasonal differences between DJF and JJA in precipitation standard deviation. In DJF, there is more precipitation variance in the Ross Ice Shelf/Victoria Land region, whereas the variance is higher along the West Antarctic coast and Amundsen Sea Embayment in JJA. The northern Antarctic Peninsula stands out as having high precipitation variance in both seasons, and the magnitude of the largest standard deviations are consistent between JJA and DJF, at maximum values of up to 300 mm w.e.. Across all

variables, Figure 2 shows that the Victoria Land coast and Ross Ice Shelf consistently stands out as a region of larger standard deviations in both seasons, indicating that there is intra-RCM disagreement about the near-surface meteorology in this location. The reasons for this will be discussed in section 4.





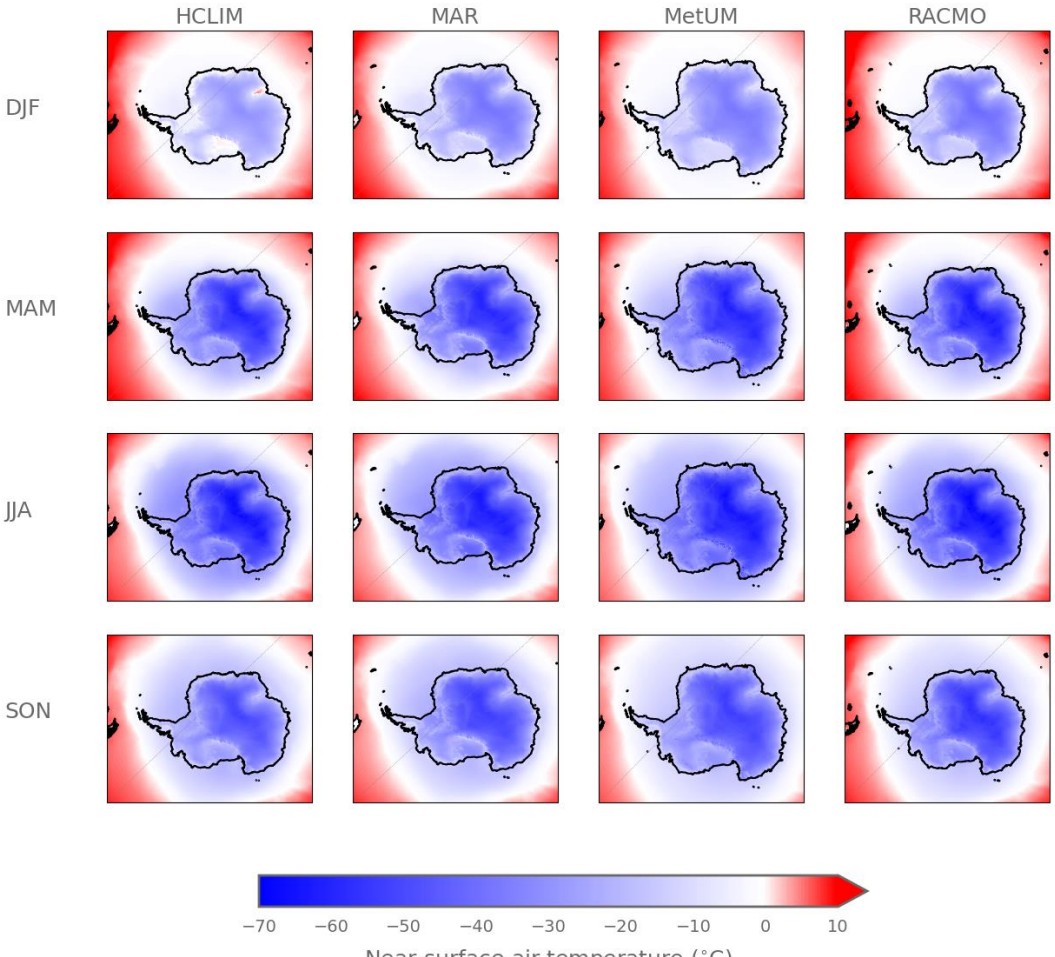

**Figure 3: Seasonal mean daily mean near-surface air temperature (in °C), during the period 2000-2019 for each of the four RCMs: HCLIM, MAR, MetUM and RACMO.**

Figure 3 shows seasonal mean daily mean near-surface air temperature for all RCMs individually. At the continental scale, the pattern of mean air temperatures is very similar between RCMs, which is consistent with all four models being forced with the same ERA5 data, and the long period over which the data are averaged. As shown in the top row of Figure 3, there is a large area of temperatures at or near the melting point throughout the summer, corresponding to the sea ice and surrounding ocean area. This contrasts with other seasons, where the only area that is at or near the melting point of sea ice is close to the sea ice edge.



As shown by Figure 3 and Table 1, the warmest RCM over the Antarctic continental area in DJF is HCLIM, with
a continent-wide mean seasonal mean air temperature of -15.6°C. The warmest RCM in JJA is MAR, with a
continent-wide mean seasonal mean air temperature of -40.3°C. Meanwhile, RACMO is the coldest RCM in the
ensemble during DJF and JJA, with a continental average seasonal mean air temperature of –23.0°C and –43.0°C,
respectively.  All RCMs are warmer on average than ERA5 in DJF, which has a mean air temperature of –23.3°C,
while in JJA all RCMs except RACMO are warmer than ERA5, which has a mean of –42.5°C (Table 1).


Figure 3 shows that there are some differences between RCMs at the sub-continental scale. For example, as also
noted above, HCLIM is much warmer over ice shelves in summer than any other model, with mean DJF daily
mean temperatures at or very close to the melting point of 0°C - this is also reflected in the much-higher continent-
wide average seasonal mean temperature of –15.6°C given in Table 1. This warmth is especially evident over the
Amery ice shelf, East Antarctica, where seasonal mean temperatures for DJF are well above zero (Fig. 2a).
However, even the large ice shelves (Filchner-Ronne and Ross) have mean summertime temperatures near 0°C
in HCLIM. These regions of particular warmth in HCLIM are also apparent in Figure S1, which shows seasonal
means of daily maximum near-surface air temperature. In DJF over the Amery Ice Shelf and the edge of the Ross
Ice Shelf, seasonal mean daily maximum temperatures are several degrees above 0°C, and there are particularly
notable differences between DJF mean and maximum air temperatures (i.e. between the top row of Figure 3 and
Figure S1). To a more limited extent there are also differences between HCLIM-simulated mean and maximum
air temperatures over the Antarctic Peninsula in all seasons.

There are other regional differences between the RCMs shown in Figure 3 too. For instance, the MetUM has much
colder temperatures in the region of steep terrain around the edge of the Ross Ice Shelf, while the other models
have warmer temperatures, which are near 0°C in some seasons. Colder mean DJF temperatures are also shown
in Figure 3 for MAR in the Weddell Sea, one of the few regions in Antarctica which commonly retains sea ice
over the summer months (Gupta et al., 2025).



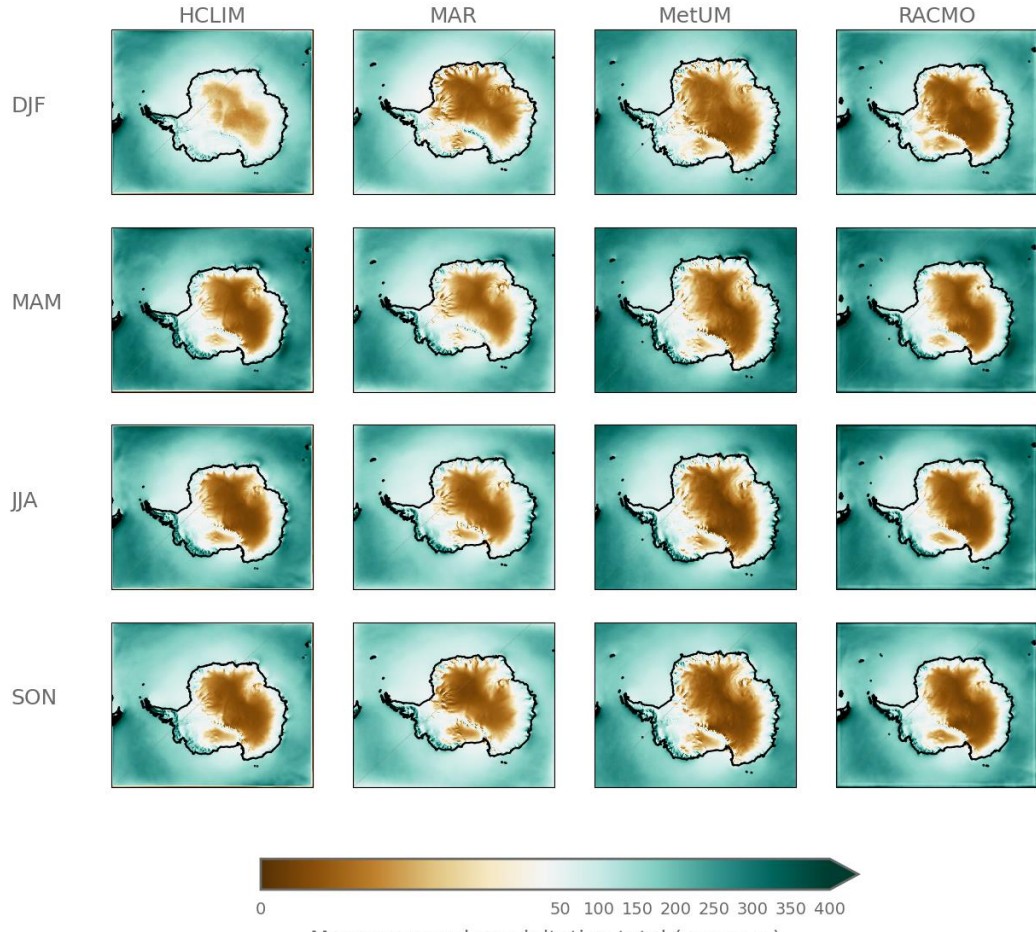

**Figure 4: As in Figure 3, except showing mean seasonal precipitation (in mm w.e.).**

As shown in Figure 4, precipitation totals in all RCMs and ERA5 and in all seasons are highest around the coastal margins and on the Antarctic Peninsula. Daily mean precipitation totals are highest in JJA, with seasonal totals in excess of 300 mm w.e.. This is consistent with the continent-averaged totals given in Table 1, which shows higher continent-averaged precipitation in JJA than DJF for all models except HCLIM. Table 1 also shows that precipitation totals are higher in the RCM ensemble than in ERA5, with RCM values ranging from 504 to 776 Gt in DJF and from 684 to 758 Gt in JJA, versus ERA5's 396 and 529 Gt for DJF and JJA, respectively.

As shown by Figure 4, in DJF HCLIM and MAR have higher daily mean precipitation totals along the edge of the Ross Ice Shelf, while RACMO and the MetUM have on average comparatively less. In all seasons, MAR has relatively less precipitation than the other RCMs over the Amundsen Sea. All RCMs show a region of high daily mean precipitation off the coast of Wilkes Land and Adelie Land. This is especially true of MAM.

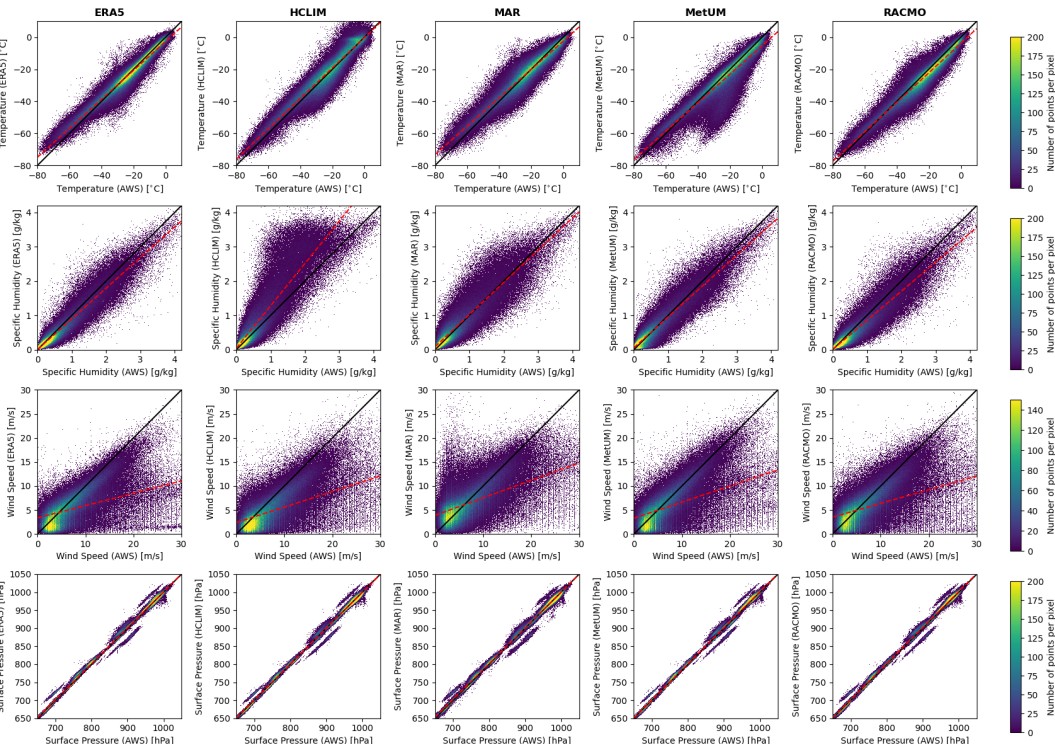

**Figure 5: Scatter plots of daily mean elevation-corrected RCM-modelled (y axis) vs observed (x axis) near-surface meteorological variables: air temperature (tas), specific humidity (huss), wind speed (sfcWind) and surface pressure (ps). Data points represent the relationship between the nearest model grid point at any given AWS location, and their colours indicate the number of points per pixel with the same model-observation values. The 1:1 line is shown in black and model linear regression fit in red.**

### 3.2 Comparison with AntAWS observations

Figure 5 shows the bias of all four RCMs and ERA5 relative to the AntAWS dataset. The difference in slope between the 1:1 line and the model linear regression, shown in each panel of Figure 5 in black and red, respectively, tells us about how closely the various models replicate the observed near-surface meteorological variables. In general, ERA5 and the four RCMs simulate air temperature and surface pressure relatively better than specific humidity and wind speed, and the RCMs mostly reproduce the observations more closely than ERA5. Air temperature, shown on the first row of Figure 5, is represented relatively well by all four RCMs, with a close agreement between the linear regression and the 1:1 line. There is more spread in the MetUM at observed temperatures between -15°C and -35°C, suggesting that the MetUM occasionally has a cold bias in some regions during periods with observed temperatures in this range. Similarly, HCLIM appears to sometimes overestimate observed air temperatures that are slightly below 0°C, which is consistent with the spatial patterns shown over ice



shelves in Figure 3. All RCMs exhibit greater variability compared to AntAWS near-surface air temperature than ERA5.


The second row of Figure 5 shows that ERA5 and all RCMs except HCLIM underestimate near-surface specific humidity compared to AntAWS. Of the RCMs, MAR reproduces the observations most closely, though with a larger spread than the MetUM, the next-closest RCM. HCLIM is the only model to overestimate specific humidity, which may also be related to the warmer temperatures shown in Figure 3.


All RCMs and ERA5 underestimate near-surface wind speeds, with considerable variability, as shown in the third row of Figure 5. Some of this is likely to do with the winds in ERA5, because there is considerable similarity between ERA5 and the RCMs. However, it must be noted that wind speed measurements are challenging to make and more prone to measurement error than other near-surface meteorological variables (Wang et al., 2023). This

is discussed in more detail in section 4.

As shown in the bottom row of Figure 5, surface pressure is represented very well in all RCMs and ERA5, with little variation between models. This is to be expected because surface pressure is a comparatively slow-changing meteorological variable which is impacted almost entirely by the large-scale atmospheric conditions. Pressure is

well constrained in reanalyses like ERA5 and these large-scale conditions are inherited by the RCMs from the forcing.





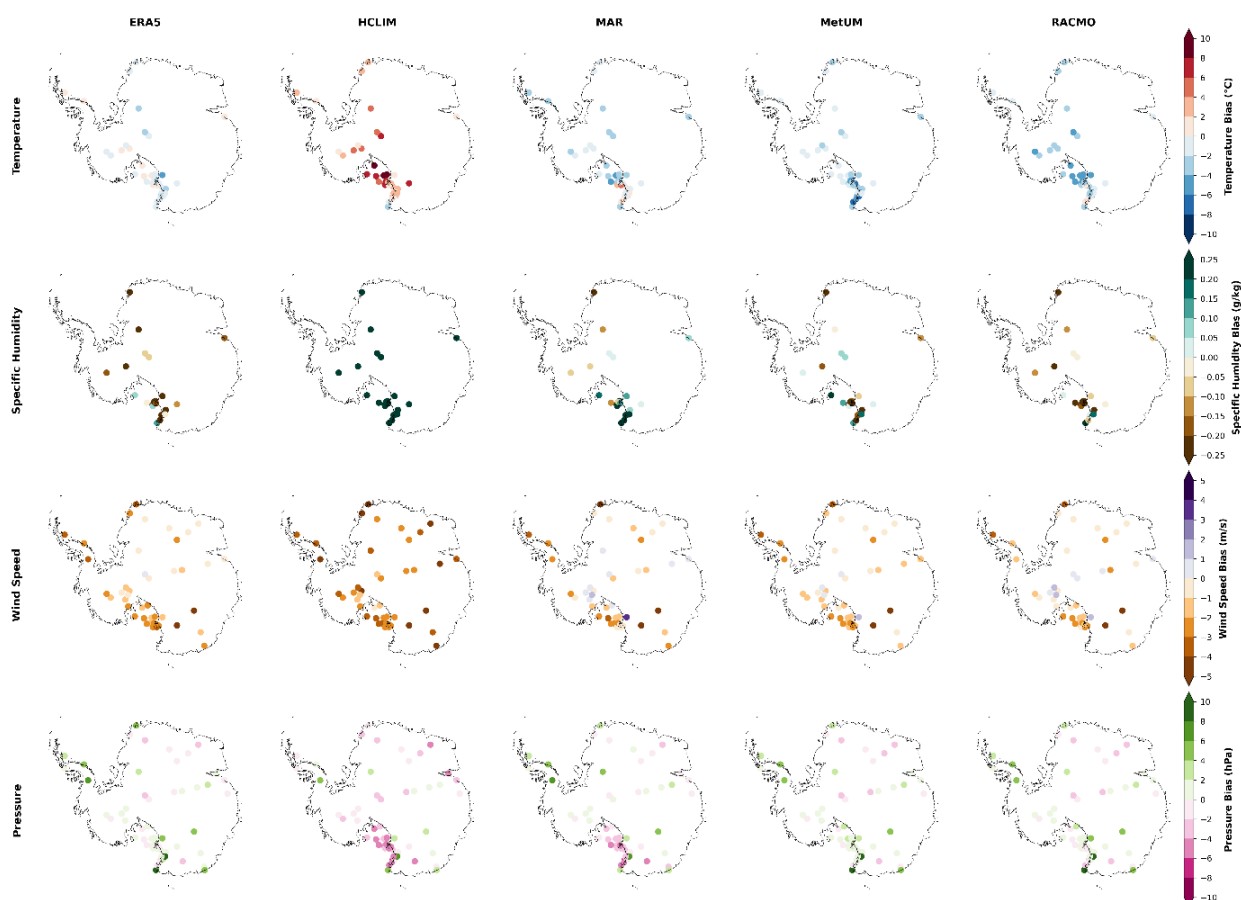

**Figure 6: Mean summer (DJF) monthly mean biases of daily elevation-corrected mean near-surface variables (rows, top to bottom): air temperature, sensible humidity, wind speed and surface pressure, all shown relative to the AntAWS dataset. Models shown are (left to right): ERA5, HCLIM, MAR, MetUM and RACMO.**



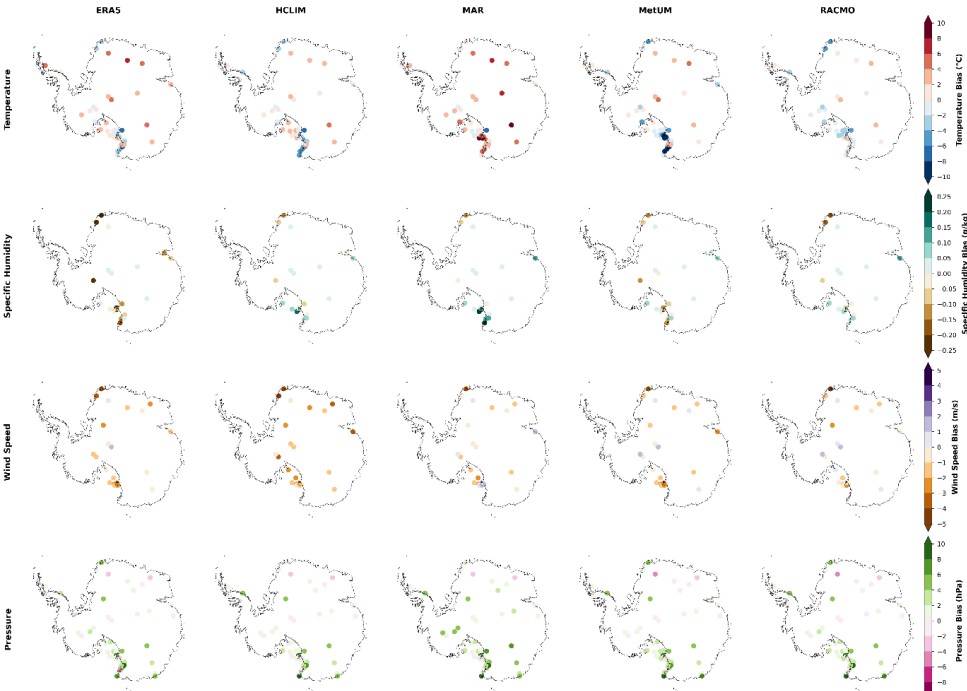

**Figure 7: As in Fig. 6, but for winter (JJA).**

We next consider the spatial distribution of biases compared to observations. Figures 6 and 7 show the mean seasonal biases of ERA5 and the four RCMs relative to the AntAWS dataset for summer (DJF, Figure 6) and winter (JJA, Figure 7) for four key near-surface meteorological variables. During summer, most of the RCMs

have a slight cold air temperature bias like ERA5, which suggests that at least some of the bias is inherited from the driving conditions. However, most RCMs have a larger bias than ERA5, implying that RCM physics are also partly responsible. The largest DJF RCM cold bias is in RACMO (-2.67°C, Figure 8), while the smallest is in MAR (-1.61°C, Figure 8). HCLIM is the only RCM to exhibit a considerable positive air temperature bias (+3.82°C), with positive values over the Ross ice shelf and nearby land ice area in Victoria Land. The picture for

specific humidity is mixed, with variation between RCMs. At most stations, the RCMs and ERA5 have a dry bias, although there is a wet bias in ERA5 along the Eastern edge of the Ross ice shelf and in Victoria Land, where the RCMs all also have large biases of mixed sign. As shown in Figure 6 and Figure 8, wind speeds are negatively biased compared to AntAWS in all RCMs (-1.34 to -2.84 m s$^{-1}$) and ERA5 (-2.27 m s$^{-1}$). Negative biases are more widely distributed and of larger magnitude than positive wind speed biases, which tend to be concentrated in the

continental interior, rather than at coasts (Figure 6). RCM wind speed biases are spatially consistent with the biases in ERA5, suggesting that much of it comes from forcing. Surface pressure biases are minimal, with continent-average biases less than 2.5 hPa for most models. ERA5, RACMO and the MetUM have the lowest biases (+0.8, 0.65 and 0.78, respectively, Figure 8), while HCLIM and MAR have larger negative biases (-2.38 and –1.23 hPa, respectively, Figure 8). As shown in Figure 6, the Ross sector once again appears as the region

with the largest biases, especially in MAR and HCLIM, which likely explains why these two models have larger Antarctic-averaged negative biases.





Comparing Figures 6 and 7 we can see that biases in JJA are mostly smaller than in DJF for temperature and humidity. As shown in Figure 7, the largest winter air temperature biases are again on the Victoria Land coast: most of the RCMs have a cold bias in this region that appears to be inherited from ERA5. The exception is MAR, which has a strong warm bias here. As also shown in Figure 8, the largest wintertime warm bias across the whole continent is in MAR (+2.0°C), while the largest cold bias is in the MetUM (-2.75°C). RACMO is consistently too cold and HCLIM too warm over the Ross ice shelf, while MAR and the MetUM have more mixed biases here (Figure 7). Specific humidity biases in all models are small, likely because winter is already extremely dry. Once again, the RCMs and ERA5 disagree on the Victoria Land coast about the direction of the bias (Figure 7). JJA surface wind biases are, like in DJF, mostly negative. They are also of small magnitude, typically below ± 2 m s$^{-1}$, with the largest negative biases in ERA5 and all RCMs in Victoria Land and Dronning Maud Land, and slight positive biases close to the South Pole in RACMO and the MetUM. In contrast to DJF, wintertime surface pressure biases are generally more positive in all RCMs, ranging from 1.32 to 2.4 hPa when averaged across the entire continent (Figure 8).

As shown in Figure 8, continent-averaged temperature biases are generally larger in DJF compared to JJA, and mostly negative. The sign of the bias switches between DJF and JJA for HCLIM, MAR and ERA5. Surface pressure biases are both negative and positive in DJF and only positive in JJA. Specific humidity biases are larger in DJF compared to JJA, especially for HCLIM, which is far too warm and wet in DJF. Across both seasons, higher elevation sites have better agreement with AntAWS than lower elevation sites, with better performance with respect to bias (Figures 6 and 7), RMSE and correlation coefficient (not shown). However, there are far fewer sites at higher elevations because most stations and AWSs are located near the coasts. This means that the values shown in Figure 8 are slightly biased towards regions with higher spatial coverage, i.e. lower elevations, including the station-dense Victoria Land region where the largest biases are found.




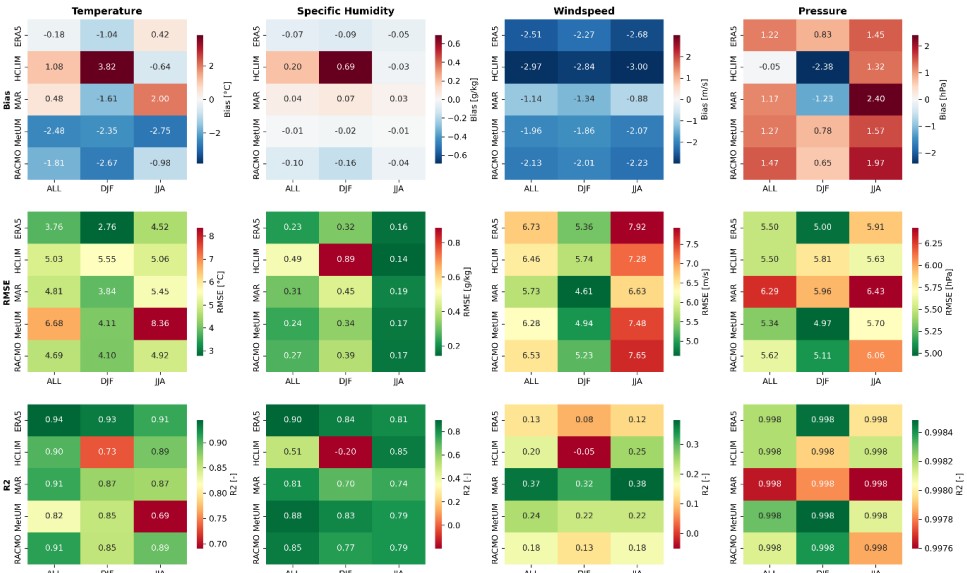


**Figure 8: Heatmap showing summary statistics for ERA5 and the RCM ensemble relative to the AntAWS dataset for all four near-surface variables evaluated: near-surface air temperature (tas, °C), specific humidity (huss, kg kg⁻¹), near-surface winds (sfcWind, m s⁻¹) and surface pressure (ps, hPa). Statistical descriptors are given, including mean bias, root mean square error (RMSE) and correlation coefficient**

**(R2). Statistics are given for the entire year in the first column, and for summer (DJF) and winter (JJA) in the second and third columns, respectively. Biases, RMSEs and R2s for each model are colour coded according to their accuracy and proximity to observations. Statistics are calculated using daily mean data from AntAWS and models.**




**Figure 9: Seasonal mean differences between each RCM and ERA5 during DJF. ERA5 data are re-gridded onto the RCM grid with bilinear interpolation to facilitate comparison.**



### 3.3 Differences between RCMs and ERA5

Figures 9 and 10 show the differences between each RCM and ERA5 in DJF and JJA, respectively. As shown in Figure 9, HCLIM is much warmer and wetter in DJF over the Antarctic continent compared to ERA5, especially over low elevations. The MetUM exhibits a similar pattern, with warmer and wetter conditions than ERA5 over land, but the magnitude of these differences is much smaller than for HCLIM. RACMO is generally colder and drier in summer than ERA5, while MAR is a mixed picture: the Victoria Land and mountainous region upstream

of the Ross Ice Shelf is warmer and wetter than in ERA5, but ice shelf areas and the sea ice area are colder than in ERA5. Temperatures are particularly cold compared to ERA5 in the Weddell Sea. All of the RCMs are cooler than ERA5 over ocean areas. Figure 9 also shows that the largest DJF wind speed differences between ERA5 and any RCM are found in HCLIM, which has lower winds than the reanalysis over most of the domain except the Ross Ice Shelf. MAR has a contrasting pattern between ocean/land, with higher wind speeds over land than ERA5,

especially on the Victoria Land coast, and slower winds than ERA5 over the ocean. Wind speeds in the MetUM and RACMO differ little from ERA5 in DJF. Generally in all RCMs the differences are larger near coasts, consistent with the improved resolution of orography in this region. Summertime differences between RCMs and ERA5 are minimal with respect to surface pressure, with the largest differences in Figure 9 over land in MAR, and over ice shelves in the MetUM. MAR, and to a lesser extent HCLIM, have more DJF precipitation than ERA5

along the coast of Victoria Land, along the eastern flank of the Ross Ice Shelf, with differences in excess of 100 mm w.e. in MAR compared to ERA5. However, MAR generally has similar precipitation totals to ERA5 over the continental land areas and much lower precipitation totals over the ocean and sea ice. By contrast, HCLIM generally has more precipitation than ERA5 over land and ice shelves in DJF, and slightly lower totals over the sea ice and ocean area near the continent, consistent with the higher temperature and humidity over land. RACMO

and the MetUM differ less visibly from the ERA5 forcing.

Some of the patterns in Figure 9 for DJF are very different during JJA (Figure 10). For example, whereas HCLIM is much warmer and wetter than ERA5 in DJF, the opposite is true in JJA, with colder and drier conditions than ERA5, particularly over ocean and sea ice areas. Figure 10 shows that the MetUM and RACMO are both cooler than ERA5 in JJA, which contrasts with the pattern shown in Figure 9 for DJF. All RCMs except the MetUM are

warmer than ERA5 in the Victoria Land coastal region during JJA, with the largest differences seen in MAR. All RCMs are drier than ERA5 over the ocean and sea ice area, except the MetUM, which is wetter across the entire domain. RACMO has a spatially consistent dry pattern compared to ERA5 in JJA, whereas differences in HCLIM and MAR are more spatially varying, with larger differences over the sea ice area. All RCMs have higher wind speeds on the Victoria Land coast than ERA5. These windier-than-ERA5 conditions are seen consistently across

land areas for MAR and across the entire domain in the MetUM, whereas HCLIM and RACMO tend to be less windy than ERA5 in JJA across most of the domain. All RCMs are consistent in their representation of winds with respect to ERA5 between DJF and JJA, with slightly larger differences in JJA for the MetUM. Surface pressure differences between the RCMs and ERA5 are more positive in JJA than DJF, with the largest differences

over the continental land area in MAR and over ice shelves in the MetUM. Precipitation differences between the RCMs and ERA5 are comparable between JJA and DJF, although differences are smaller in JJA. As shown in Figure 10, MAR has the largest (negative) precipitation difference compared to ERA5 in JJA, over ocean and sea ice areas.





**Figure 10: As in Fig. 9, but for JJA.**





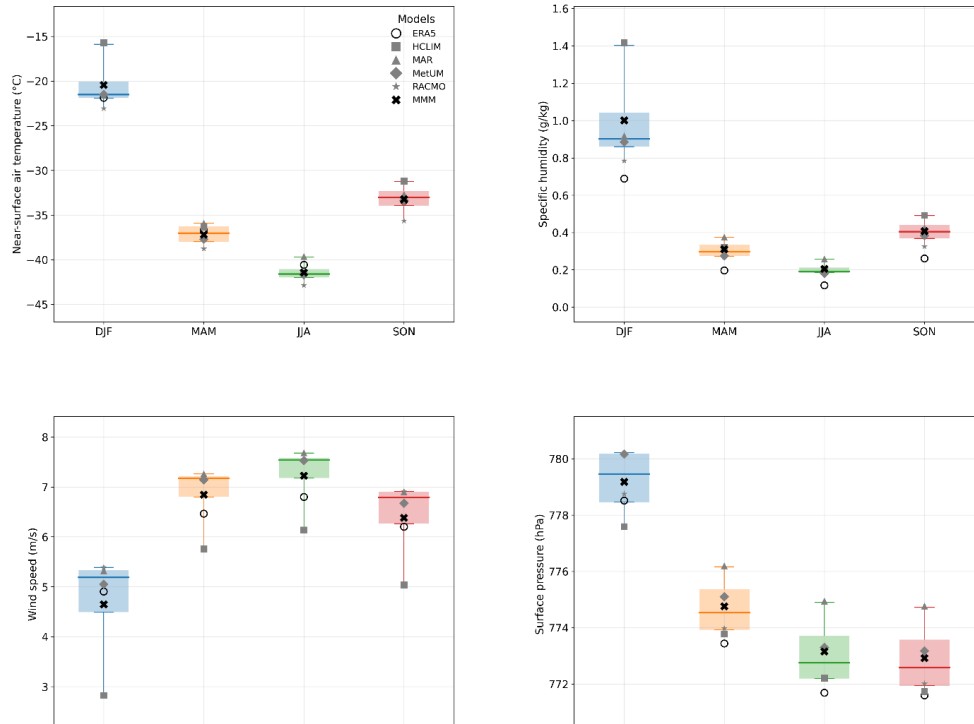

**Figure 11: Boxplot showing seasonal statistics for the four key climate variables examined in this study (near-surface air temperature (°C), specific humidity (g kg⁻¹), wind speed (m s⁻¹) and surface pressure (hPa). The grey markers indicate the mean of each RCM in the ensemble, calculated as the mean value for the specified season, averaged over the continental land area. The white circle with black outline shows the equivalent value for ERA5. The mean value of the multi-model mean (MMM) of the RCM ensemble is shown as a black cross. The coloured boxes cover the inter-quartile range (25th to 75th percentiles) of the ensemble multi-model mean, with the whiskers extending to the 1st and 99th percentiles and the median given as a solid colour line.**



Figure 11 gives an indication of the spread of the ensemble and its performance relative to ERA5, separated by season and averaged over the whole continent. For some variables and some seasons, the RCM ensemble mean is quite similar to the ERA5 mean, while for others there is more divergence between them. For instance, the RCMs

simulate very similar near-surface air temperatures to ERA5 in all seasons, with larger differences of ~1-2°C between the MMM and ERA5 in DJF and JJA, where ERA5 temperatures fall outside the MMM 75th percentile in JJA. Conversely, considering specific humidity, the RCM ensemble is systematically wetter than ERA5 by ~0.1-0.3 g kg⁻¹ with ERA5 mean specific humidity falling below the MMM 1st percentile in all seasons. Similarly, ERA5 surface pressure is below the ensemble mean by ~1-2 hPa in all seasons and outside the range of values

simulated by all the RCMs, except in DJF. Mean wind speeds in ERA5 are also consistently lower than in the RCMs by ~0.1-0.3 m s⁻¹ (with the exception of HCLIM); this is also clear from Figures 9 and 10. The greatest ensemble spread is found in DJF for all variables, driven in most part by HCLIM, whose simulated near-surface air temperature and specific humidity are above the 99th percentile for the ensemble, and whose wind speeds and surface pressure are below the 1st percentile. This is consistent with the temperature and humidity differences shown over land ice in Figures 8 and 9 in DJF. The Antarctic-wide HCLIM mean is an outlier in DJF for all

shown over land ice in Figures 8 and 9 in DJF. The Antarctic-wide HCLIM mean is an outlier in DJF for all



variables, and HCLIM's wind speeds are lower in all seasons than both the ensemble MMM mean and ERA5 mean. Wind speed is the variable with proportionally the largest inter-quartile range relative to its mean, indicative of larger inter-model variability, as is also shown by Figure 5. The lowest inter-quartile range is found in near-surface air temperature and surface pressure in all seasons, particularly JJA. This indicates that the RCM ensemble

has good inter-model agreement on these variables at the continental and seasonal scales.

**4 Discussion**

The RCMs shown here offer an improvement over ERA5 due to their higher resolution and more sophisticated physical parameterisations. This is particularly important around the coasts, where orography is more realistically represented at 11 km grid spacing compared to the ~31 km of ERA5 or the 50-100 km scale of most global climate

models. As shown in Figures 9 and 10, this results in the largest differences in terrain-sensitive variables like winds, surface pressure and precipitation around the coastal margins of the continent. This is also clear at the continental scale, as shown in Figure 11, which demonstrates that the RCM ensemble has consistently higher wind speeds and surface pressure on average than ERA5, indicating that higher resolution orography affects the representation of the near-surface meteorology. However, while this kind of resolution provides a marked

improvement, especially in the representation of extreme events, localised weather features and terrain-sensitive variables, several studies have found that even the 11 km resolution here is insufficient to capture the full range of observed phenomena (Gilbert et al., 2020; 2022b; 2025; Zentek & Heinemann, 2020; Datta et al., 2019; Laffin et al., 2021). For instance, Gilbert et al. (2025) find that resolutions of 12 km – comparable to the resolution of this RCM ensemble – are sufficient to capture total precipitation during extremes, but that resolutions of ~1 km

are required to represent phase partitioning between rain and snow. Similarly, other studies show that resolutions that are impractically high to run for the duration of this ensemble (e.g. 1-5 km grid spacing) are required to capture the full range of complexity with respect to cloud microphysics (Gilbert et al., 2020) foehn winds (Datta et al., 2019; Gilbert et al., 2022b; Elvidge et al., 2020; Orr et al., 2021), and extreme events (Gutowski Jr. et al., 2020; Gilbert et al., 2025). However, on the climatological scale, regional features like the persistent coastal low-

level easterly winds are comparatively insensitive to resolution (Caton Harrison et al., 2025).

The benefit of an RCM ensemble is that the differences in model physics, parameterisations, and implementation methodologies can help explore the full range of uncertainty associated with a given forcing. The ensemble was designed to maximise the usefulness of these differences to understand the processes and features of the Antarctic

climate. By running all the RCMs at a horizontal resolution of ~11 km over the same domain and with the same forcing, we eliminate uncertainty associated with starting conditions or forcing data, allowing us to harness the strengths of improved RCM physics to understand Antarctica's climate. Importantly, the method used to constrain the RCMs to the forcing data does not appear to impact their capacity to reproduce the observed climate. Specifically, the MetUM, the only model to use a re-initialisation approach rather than a nudging one, exhibits no

consistent noticeable differences from the other RCMs, which all implement traditional relaxation-based nudging, albeit at different time intervals and locations for some variables. No single RCM stands out as better or worse overall than any other, demonstrating the value of using an ensemble that has different strengths in different aspects. As we have demonstrated, the ensemble represents observed conditions more closely than ERA5, suggesting that this new dataset is a state-of-the-art resource that can be used for scientific studies of Antarctic

weather and climate over the recent past.



A clear limitation of this study is that we could not evaluate the ensemble's performance with respect to precipitation. There are no available datasets of total precipitation that have sufficient spatial or temporal coverage, or sufficiently low uncertainty to conduct a thorough and reliable comparison. While this is possible on a case study basis (as done by e.g. Gehring et al., 2022; Simon et al., 2024; Gilbert et al., 2025), it is not possible over the long (climatological) time period we consider here. Satellite products have limitations, including poor coverage over the poles and biases associated with clouds and atmospheric conditions. Moreover, satellite products rely on models to convert raw signals in the form of reflectance or polarisation into meaningful values, meaning the comparison is not against a true 'observation'. However, we have included RCM mean values and the comparison against ERA5 here, in order to present useful information about Antarctic precipitation. In the absence of widespread and long-running observations of snow and rain, the PolarRES ensemble is another useful dataset with which to evaluate Antarctic precipitation over the recent past.

We were able to validate the rest of the variables against in situ observations, however. The RCMs all perform fairly well with respect to the AntAWS dataset, which may be at least partially influenced by the fact that some AntAWS stations are assimilated into ERA5. However, as shown in Figures 5, 6 and 7, wind speed is the variable with the highest biases across the whole ensemble and ERA5. It must be noted that wind speed observations are difficult to make and are typically the least reliable variable of the suite usually measured by AWSs (Wang et al., 2023). Errors in the recorded height of the sensor can easily occur, especially in regions with considerable snowfall, which buries the AWS and changes the height of the sensor over the course of the season (Genthon et al., 2021; Wang et al., 2023). This can be corrected if snow height sensors are also available, but this is not currently possible with the AntAWS dataset. Blowing snow can also cause highly localised accumulation, which can impact wind speed and direction observations, and high winds can cause the towers used to sample the atmosphere to tilt, therefore also affecting measurement height. Because winds vary strongly in the first few metres of the Antarctic atmosphere, this can introduce bias and noise in the dataset, which is reflected in the wide spread of wind speed values in Figures 5 and 11. These observation challenges may make the comparison against model data less coherent and therefore reduce the correlation between observations and model outputs at any given grid point (Figures 6 and 7). At the scale of the simulations (~11 km), the RCMs may also not be capturing localised wind features that occur at sub-grid scale.

Comparing summer to winter, there are no clear-cut improvements in one season relative to the other. While it appears from Figures 6 and 7 that overall biases are smaller in JJA for some variables and models, this can be explained by the extremely cold, dry conditions in winter that mean even small temperature and humidity biases can represent a large percentage-wise deviation from observations. Surface pressure biases are small in both seasons and (coastal) wind speed biases are relatively large in both seasons, as discussed above. Moreover, RMSEs can be high and correlations with observations low, even when biases are fairly small, as shown in Figure 8. We must also state that the comparison of summer vs winter is not an entirely fair one, because the distribution of AntAWS stations is not the same between the seasons for all variables. Due to the presence of data gaps in some seasons and the application of our filtering criteria (described in section 2), certain stations are excluded for one or the other season for some variables resulting in different coverage between JJA and DJF. For instance, East Antarctica has more station observations of near-surface air temperature in JJA than DJF, whereas the spatial coverage is better in DJF for all other variables.



The Victoria Land region along the eastern edge of the Ross Ice Shelf stands out in the dataset as a region that the
RCMs and ERA5 struggle with. This may be partially explained by the different ways that the models handle land
surface type. This is one of the very few regions in Antarctica that is not entirely ice-covered, and different models
have different approaches to handling this, particularly at this sort of resolution where grid cells may contain a
mixture of surface types. For instance, in some RCMs this can be represented with a land/ice fraction (MAR,
RACMO), while in others the land surface is specified as simple bare ground, bare ice or snow-covered ice
(HCLIM, MetUM). These differences can generate biases of varying sign in near-surface variables like
temperature and specific humidity because the land surface type impacts the albedo and surface energy fluxes to
and from the surface. Moreover, this region is also extremely steep, and the complex terrain can generate a variety
of wind effects, including foehn winds, katabatic winds, and features like the Ross Air Stream (Seefeldt &
Cassano, 2012), which may partly explain some of the wind speed biases in the Victoria Land region. Some RCMs
may be simulating more katabatic outflow (and hence colder temperatures) near this steep terrain (e.g. the
MetUM), whereas the other RCMs may be simulating more compressive warming such as foehn winds, or
adiabatic compression of katabatic flow.

RCM-specific biases and features can largely be explained by idiosyncrasies in important parameterisation
schemes. For example, as shown in Figures 6 and 9, HCLIM is much warmer and wetter in summer than both
observations and ERA5, respectively. This is likely due to the way that HCLIM parameterises snow albedo as a
function of snow grain size. HCLIM has not been used extensively in Antarctica before, and the default scheme
assumes snow grain size as a function of snow density, following an empirical relationship from Anderson (2008)
based on measurements from the NOAA-ARS snow research station in Danville, Vermont (USA). In polar
conditions, snow grain sizes are typically smaller (Libois et al., 2015). Overestimating snow grain size leads to
underestimated snow albedo, resulting in excessive shortwave absorption at the surface which affects the
morphology of the snowpack and the development of its albedo in high-melt conditions such as those seen over
ice shelves in DJF. These biases in albedo translate into much warmer conditions than observed, which in turn
produces a humidity bias because warmer air has a higher specific heat capacity. The extra warmth and moisture
in the HCLIM atmosphere can therefore also at least partly explain the higher summer precipitation totals in
Figure 9 compared to ERA5. Verro et al. (2025) document this issue, showing surface temperature biases
exceeding +5 °C over sea ice when using the default scheme. As a result of the work in that study, the albedo over
sea ice - but not over land - has been corrected in the model version used here, which means there is no
considerable warm and/or moist bias over sea ice. A comparable adjustment to the scheme over land ice is
necessary to reduce biases over ice shelves. MAR exhibits different behaviour to the other RCMs over the sea ice
area, as shown in Figures 9 and 10. MAR simulates colder conditions over the sea ice area in both DJF and JJA,
which may be related to its snow-on-sea ice scheme. The MetUM has a positive surface pressure bias over ice
shelves because the elevation of ice shelves at the calving front is given as 0 m, rather than the typical 50-200 m
that are observed above the water level in reality. This results in higher surface pressure values compared to ERA5,
as shown in Figures 9 and 10. Figures 9 and 10 also show that the MetUM also tends to be more humid than other
RCMs in cold environments, something also shown by Gilbert et al. (2025).



## 5 Conclusions

We have shown that the PolarRES ensemble offers an improvement in the simulation of the Antarctic near-surface atmosphere compared to ERA5. The RCMs are capable of better capturing the range of values measured in the

AntAWS dataset during the period 2000-2019, and offer a finer scale representation of the Antarctic climate over the recent past. These improvements relative to reanalysis and previous RCM datasets produced under Polar CORDEX are partly associated with the enhanced resolution, especially in regions with complex terrain such as the Antarctic Peninsula and near the coastal periphery. They are also impacted by recent changes made to the RCMs to improve their ability to simulate this complex polar environment. This ensemble includes the latest

release of RACMO2.4, and is the first time that HCLIM has been used over a pan-Antarctic domain. These physics changes, plus the addition of an entirely new and capable RCM, offer substantive improvements for our ability to understand the Antarctic environment. In a region where observations are still sparse and have large observational uncertainty, RCMs are an invaluable analytical tool. This ensemble therefore represents a state-of-the-art resource for examining weather and climate processes and patterns in Antarctica.


This study also underlines the need for more observations of Antarctic near-surface meteorology, particularly in regions with poor spatial coverage, and of variables with relatively fewer measurements, like precipitation. Precipitation is still difficult to evaluate, and indeed we were unable to validate the ensemble against observations in this study because in situ observations are so spatially and temporally sparse and prone to uncertainty. It has

already been noted in the literature, e.g. Gilbert et al. (2025) that more in situ precipitation observations are desperately required for model evaluation and process studies. Installing precipitation sensors, particularly those capable of segregating by phase, should be prioritised by funding agencies and national Antarctic programmes to advance our ability to simulate, forecast and understand the polar climate.

The data from the PolarRES ensemble are freely available on the Earth System Grid Federation and we encourage researchers to make use of this new and exciting dataset for their own scientific applications. Given the strengths and limitations of the various RCMs in different domains and aspects, we stress that careful consideration and case-specific evaluation should be undertaken by users to verify that the RCM(s) that they have chosen for their research questions are appropriate. By identifying these strengths and limitations, we have highlighted priority

areas for ongoing model development that will continue to improve these RCMs' ability to simulate the polar climate. It was beyond the scope of this work to evaluate the performance of the ensemble with respect to the full range of variables outputted by the RCMs, but forthcoming publications will focus on their simulation of particular elements of the Antarctic climate system such as surface mass balance, energy budgets, extreme events and sea ice.


RCMs are increasingly used to downscale global climate projections from earth system models, and the PolarRES ensemble has been used to project storylines of future Antarctic climate change out to 2100 using representative global climate models identified by Williams et al. (2024). These simulations will capture a range of projected future climate outcomes for the Antarctic, and the results of analyses will be presented in forthcoming studies.

The evaluation presented here demonstrates that these RCMs are capable of representing the contemporary climate and indicates that they will offer useful insight into projected future changes in this important and rapidly changing region.



**Author contributions**

EG, MH and JAT contributed equally to leading this work. EG led the writing and analysis, and developed, produced, and post-processed the MetUM model output with the help of AO, SG and TP. JAT and MH contributed substantially to the writing, visualisation and analysis work, producing many of the figures in this paper. JAT also developed, produced and post-processed the HCLIM model outputs, with help from KV, OBC, FB, MO, NH and RM. DM, XF and CL developed the MAR model and worked on the post-processing and data delivery. WJB, CD, KV and MT developed RACMO and produced and post-processed its outputs. PM, RM, WJB, AO and XF

contributed to project management and funding acquisition. All authors reviewed and commented on the manuscript.

**Competing interests**

The authors declare that Ruth Mottram and Xavier Fettweis are editors of *The Cryosphere*.

**Acknowledgements**

Huge thanks to technical support staff at each modelling institute who helped in the production, post-processing and sharing of the RCM data. Thanks to Laura Gerrish at the British Antarctic Survey's Mapping and Geographic Information Centre for producing Figure 1.

**Funding**

The authors acknowledge funding from PolarRES. The PolarRES project received funding from the European
Union's Horizon 2020 research and innovation programme call H2020-LC-CLA-2018-2019-2020 under grant agreement number 101003590. Nicolaj Hansen gratefully acknowledges financial contributions by the Danish State through the National Centre for Climate Research (NCKF) and the Novo Nordisk Foundation project PRECISE (grant no. NNF23OC0081251)

**Data availability**

The PolarRES ensemble data will be freely available via the Earth System Grid Federation (ESGF) system shortly. In the interim and for the purposes of review the data can be found at:
MAR: http://ftp.climato.be/fettweis/MARv3.13/PolarRES/Antarctic/MAR-ERA5/
MetUM: https://gws-access.jasmin.ac.uk/public/polarres/MetUM_PolarRES/Antarctic/
HCLIM: https://download.dmi.dk/Research_Projects/polarres/ANTARCTICA/
RACMO: https://doi.org/10.5281/zenodo.16902106



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
