# Peer review of "The PolarRES dataset: a state-of-the-art regional climate model ensemble for understanding Antarctic climate"

_EGUsphere, 2025_

## Author Comment (AC2)

**Author response to Referee #1**

General Comment:

This manuscript presents and evaluates a new, high-resolution (~11 km) regional climate model (RCM) dataset developed through the PolarRES project. The ensemble combines four advanced RCMs (HCLIM, MAR, MetUM, and RACMO) to simulate Antarctic weather and climate for 2000–2019, providing a much finer-scale view than reanalyses such as ERA5 and improving the representation of key features like coastal winds, precipitation, and temperature gradients shaped by the continent's complex terrain.

Methodologically, the paper advances Antarctic climate research by introducing a consistent multi-model ensemble framework that enables systematic comparison among models with differing physics. All simulations are forced with the same ERA5 reanalysis data, isolating differences in model behaviour rather than boundary effects. Validation against the continent-wide AntAWS observational network further strengthens confidence in the ensemble's ability to reproduce near-surface climate processes at high spatial detail.

The manuscript includes one table summarising mean near-surface climate statistics and eleven main figures, supplemented by additional material in the Supporting Information. This clear and data-rich structure effectively supports the manuscript's evaluation of the PolarRES ensemble.

We would like to thank the referee for their comments and we address them here. In black are given the comments, in blue our response and in orange the changes we would implement in the manuscript.

Specific comments:

The following feedback is provided to the authors for their consideration to help improve the manuscript, should it be considered for publication:

It would be useful for readers if the timespan of the comparison (2000-2019) was noted in the abstract.

We have now included this: "Here we present and evaluate a comprehensive, high-quality, ~11 km resolution RCM dataset, the PolarRES ensemble, for the period 2000–2019."

Unless there is another PolarRES consortium paper published with technical details of each of the RCMs (including their domains) more information about these will need to be

provided in Section 2.2. In particular, the domain sizes will be of interest to readers and the extent of the domains surrounding Antarctica.

We thank the reviewer for this comment. A dedicated PolarRES consortium manuscript describing the full modelling protocol, including technical details of each RCM and their respective domains, is currently in preparation. We are therefore attaching a draft version of this manuscript for the reviewer's reference. This manuscript is planned for submission in February, and we kindly ask that this document not be shared at this stage. In the revised manuscript, we will cite this PolarRES protocol paper.

Further information about the AntAWS wind speed data would be of interest. It is noted on L192-194 that a correction was applied to resolve differences in surface elevation and air temperature, but how are observed near-surface wind speeds and the lowest modelled wind speeds reconciled? This is relevant to the wind speed data presented in Figure 2, Table 1 and other figures presenting wind speed information.

The observed 3m wind speed was indeed recalculated to 10m wind speed, this is now clarified in the methods as follows: "Besides, the observed wind speed measured at 3 m height is recalculated to 10 m assuming a logarithmic wind profile, enabling a direct comparison with the modelled 10 m wind fields."

The comment that the pattern of mean air temperatures "is very similar" on L273 could be reworded, as this is not entirely true for summer in the HCLIM simulations. Is there any explanation for the warm bias in all models in summer, as noted on L284?

We have rephrased 'is very similar' to 'is broadly similar' and have added nuance to highlight that HCLIM is quite different in summer: "However, notable inter-model differences remain. In particular, HCLIM stands out during summer, showing warmer near-surface air temperatures compared to the other RCMs."

"This is probably a twofold problem. Firstly, HCLIM-ALADIN simulations produce enhanced cloud ice content, which leads to increased cloud optical thickness. Consequently, downward longwave radiation at the surface is amplified, resulting in a warm near-surface temperature bias over the Antarctic continent (Kolbe et al., 2025). Secondly, HCLIM snow albedo is not yet tuned for Polar conditions. The default scheme assumes snow grain size as a function of snow density, following an empirical relationship from Anderson (2008) based on measurements from the NOAA-ARS snow research station in Danville, Vermont (USA). In polar conditions, snow grain sizes are typically smaller (Libois et al., 2015). Verro et al. (2025) document this issue, showing surface temperature

biases exceeding +5 °C over sea ice when using the default snow scheme. The biases in clouds and in snow albedo translate into warmer conditions than observed, which in turn produces a humidity bias because warmer air has a higher specific heat capacity. Cloud and albedo biases in climate simulations are linked; however, distinct issues have been identified for each. Evaluation and improvement of the cloud microphysics, and the snow scheme of the model are ongoing at the time of publication."

In relation to surface winds, if point 3 is clearly addressed, then the paragraph starting on L346 might need to be refined, as the height differences between observations and the lowest level of modelled wind may explain some of the differences.

We did correct for the height difference between AWS (3m) and RCMS (10m), therefore this difference cannot explain the differences we find. This correction is now explicitly mentioned in the methods section after point 3. Other errors in the measurements that could lead to this discrepancy are mentioned in the discussion (eg errors in observation height due to snow burying sensors)

It would be of interest to readers if further insights as to why the RCMs are cooler than ERA5 over ocean areas could be provided (L441-442). This cold bias could have important implications for those using the data for ocean-to-land interactions, so any insights about this would be valuable.

We have rephrased the sentence, as the bias is mostly present over sea ice cover and is much smaller over open ocean. This part reads now as "Temperatures are particularly cold compared to ERA5 in the Weddell Sea, especially in MAR. This is likely related to the presence of compact sea-ice cover, which is represented by a different surface scheme than in ERA5."

Is the comment "no single RCM stands out as better or worse overall than the other" on L536 really an accurate reflection of the results presented? The warm and moisture bias in HCLIM would seem like a red flag at this time. An explanation is provided for this in the paragraph starting on L599. In the conclusions, it is indicated that this work has provided insights on the strengths and limitations of the RCMs, and highlighted priority areas for ongoing model development (L649-651). To make this clearer, I wonder whether an additional table in the discussion that shows current significant strengths and weaknesses might be a useful way to provide a summary to readers. If this is problematic, any attempt to strengthen your insights in the discussion and conclusions would be really useful to readers, who will be looking for quick tips or pros and cons about each RCM before using the data.

We agree that the original phrasing may give the impression that all RCMs perform equally well, which is not fully supported by the results. We therefore rephrased this statement around L536 of the original MS to reflect more strength and weaknesses of the RCMs, in particular the warm and moist bias in HCLIM. We redirect here to Figure 8 which gives an overview of the model performance. The goal of this overview is not to rank the RCMs, but to transparently summarize their relative performance for the variables and seasons, allowing users to make informed choices.

"Figure 8 provides an overview of the performance of the RCMs across variables and seasons and can serve as a practical reference for users of the dataset. While no single RCM performs best across all variables and seasons, HCLIM exhibits lower overall performance than the other three models, which have been more extensively applied and developed for Antarctic conditions. By contrast, RACMO, MAR, and MetUM show more balanced performance. MetUM performs best in representing AWS observations but has pressure biases over the ice shelves. Temperature biases are largest in HCLIM, while RACMO and MAR show a cold bias, particularly in summer, and MAR a warm bias in winter."

In relation to point 7, the comment that the RCMs all perform "fairly well" could be clearer and a stronger position one way or the other would be useful to readers (L556). In the text that follows, the issue of wind sensor height is raised, which could come earlier in the methodology.

We thank the reviewer for this comment, we have modified the text.

Lastly, it would be great if the resolution of the figures could be improved when this work is published. The figures are of a high-quality but the resolution of them is low in the preprint. If the figures are being revisited, any increase in font size of the x and y-axes and their labels to improve readability is normally welcomed by "older" readers.

The font size of the axis labels have been increased to improve readability. All figures will be saved and submitted in higher resolution for the final published version.

**Author response to Referee #2**

I have a question about EGUsphere. According to my search online it said: "EGUsphere is not a journal; it is an online platform and preprint repository for Earth, space, and planetary sciences, hosted by the European Geosciences Union (EGU). It serves as a central hub for conference abstracts, presentations, and preprints that can undergo public peer review and be submitted for publication in one of the EGU's 19 open-access journals." So which open access journal is being targeted for this paper?

We would like to thank the referee for their comments and we address them here. This manuscript is submitted to The Cryosphere. In black are given the comments, in blue our response and in orange the changes we would implement in the manuscript.

General Comments:

The authors have undertaken a massive task to derive, assemble, and evaluate the PolarRes ensemble. Potentially this data set along with its individual members at 11 km spatial resolution will be valuable for exploring aspects of Antarctica's weather and climate, especially broadscale surface mass balance. It is unfortunate that the full technical details are not available along with this manuscript. For most applications, regional model simulations specifically designed for the intended application are needed.

My comments address the assertion as to whether ensemble actually advances capabilities beyond that provided by ERA5 (31 km horizontal resolution) and by implication the soon to be ERA6 (18 km horizontal resolution). Also, I have many comments regarding the scientific content and the presentation of results. In general, a lot more use of Supplementary material should be made to provide a more refined analysis of the RCM's performance, even though a follow-on in-depth publication is planned.

The first major issue is HCLIM. The performance of this model is really deficient (e.g., Figs. 8 and 9) and significantly penalizes the ensemble. The explanation for the huge summer warm bias (and moist bias) that is especially prevalent on the major ice shelves is attributed to the deficient albedo treatment. This issue has been documented previously by Xue et al. (2022; https://doi.org/10.1007/s11707-022-0971-8) and the impact of its removal demonstrated. I wonder if this is the complete story. Are you sure that the ice shelves are not treated as sea ice? That can happen because the Antarctic coastline is specified in many terrain data sets at the edge of land ice. This situation would provide a large heat and moisture source for the atmosphere at high latitudes in summer. In winter,

the essentially 100% concentration sea ice would act very like land ice. Also, the wind speeds are much slower than the observations and other models throughout the year. Although the authors are reluctant, this model should be removed from the ensemble, or an ensemble with 4 versus 3 members should be provided. On page 27, details on the PolarRes ensemble are missing.

Thank you for this comment. We agree that HCLIM currently shows weaker performance relative to the other RCMs in this evaluation. Nevertheless, we have chosen to retain HCLIM in the ensemble because one of the objectives of this study is to introduce and document the full set of regional climate simulations produced within the PolarRES project. Including HCLIM allows us to assess how a newer and less extensively applied model behaves within a multi-model Antarctic ensemble, alongside more established RCMs. This provides valuable context for interpreting ensemble spread, highlights areas where further model development and tuning are needed and supports transparency and reproducibility for future PolarRES-based studies.

To further adress this we have added a similar figure to Figure 2 in Appendix A, in which the MMM includes only MAR, MetUM and RACMO. This allows the reader to see the impact of HCLIM in the multi-model mean and standard deviation.

Use of Victoria Land as a geographic label. It extends from 70.5S to 78S (or the latitude of McMurdo Station) and not farther south along the Transantarctic Mountains (TAM). Refined use of this term is required as it is often used to describe the entire span of the TAM.

We have corrected the use of Victoria Land as a geographic label throughout the manuscript. For example, at line 247 we revised the text to read: "The Trans-Antarctic Mountains and Victoria Land region stand out even more during JJA." Similar changes were made wherever features extending across the broader Trans-Antarctic Mountains were previously described using the term Victoria Land.

The authors need to state the difference between ERA5 and the RCMs employed.  ERA5 is the result of a very short-term forecast with the frequent assimilation of surface pressure observations. The RCMs result from much more extended forecasts that have frequent restarts (MetUM) or are nudged toward ERA5; no direct assimilation of observations occurs. I suspect ERA5 and the RCMs have very similar daily sea ice and SST forcing.

To accommodate this we have rephrased L120 to "ERA5 is generated through a sequence of short-range forecasts in which a wide range of satellite and in-situ observations are

frequently assimilated using 4D-Var within cycle 41r2 of the ECMWF Integrated Forecast System (IFS) (Hersbach et al., 2020)."

Also L123 is rephrased to "The RCMs are forced with ERA5 reanalysis, described above, but are run as extended forecasts that do not assimilate observations directly. Instead, they are either frequently re-initialised (MetUM) or nudged toward ERA5 at the lateral boundaries to constrain the large-scale circulation, and at the surface through prescribed sea surface temperature and sea ice fields from ERA5."

Line 43: "the dearth of observations on this extreme continent". More observations are needed but the situation has improved with extensive AWS deployments and many satellite data sets. I would also say that limited understanding and deficient model physics are significant contributing factors, partly related to limited relevant process observations.

We have given a more nuanced statement now: "However, although it is vitally important to understand the present-day climate of Antarctica, substantial uncertainty remains regarding its current state and future evolution. Although observational coverage has improved in recent decades through satellite products and automatic weather station networks, in situ observations remain sparse, limiting process understanding and contributing to deficiencies in model physics, particularly for complex atmospheric processes that operate at fine scales."

Line 54: The definitive work on foehn winds on the eastern Ross Ice Shelf was by Zou et al. (2019; doi: 10.1002/qj.34600).

Thanks, we have added a citation to this work here.

Lines 54-55: "Meanwhile, katabatic winds exert a dominant influence on the climate of coastal Antarctica (Heinemann et al., 2019; 2021; Parish & Bromwich, 2003; Caton Harrison et al., 2024)". The authors may mean Parish and Bromwich (2007; Mon. Wea. Rev., 135, 1961-1973.) that demonstrates the katabatic (surface) wind impact on the weather and climate of large portions of the middle and high latitudes of the Southern Hemisphere (there are some debates about the katabatic wind phraseology but this is used here generically). Foehn winds are important climatic features but not of the same impact as katabatic winds, lines 49-50.

We have corrected the citation and have rephrased this section, to first mention katabatic winds having large scale impacts and then foehn winds having more localized impacts:

"Katabatic winds exert a dominant influence on the climate of coastal Antarctica and large parts of the Southern Hemisphere (Parish & Bromwich, 2007; Heinemann et al., 2019, 2021; Caton Harrison et al., 2024), affecting near-surface climate (Renfrew & Anderson, 2003, van Wessem et al., 2015; Gilbert et al., 2022b), and the redistribution of snow through drifting snow (Gadde et. al., 2024). Foehn winds, while more spatially localised, have been shown to impact the local climate over the Antarctic Peninsula (Datta et al., 2019; Gilbert et al., 2022a; 2022b; Wille et al., 2022), Amundsen Sea Embayment (Francis et al., 2023; Gilbert et al., 2025), Ross Ice Shelf (Zou et al., 2019; Hansen et al., 2024), McMurdo Dry Valleys (Speirs et al., 2010; Hofsteenge et al., 2022) and East Antarctica (Wille et al., 2024)."

Line 76: Datta et al (2023) is not in the reference list.

This reference is now added to the reference list.

Lines 87-88: Campbell et al. (2024) examined New Zealand and Iles et al. (2020) considered Europe.

We have clarified this as follows "RCMs have also been shown to better capture extreme events, including extreme precipitation, extreme heat, atmospheric rivers, winds, and melt events, both in mid-latitudes (Campbell et al., 2024; Iles et al., 2020) and in Antarctica (Gilbert et al., 2025; Kolbe et al., 2025)."

RCM Descriptions: Please provide more details as to the nudging used by HCLIM and RACMO as this is very important to keep the model state close to reality. Levels? Frequency? Variables?

We have expanded the description for RACMO as follows:
"At the lateral boundaries, RACMO is forced with 3-hourly ERA5 data for wind vectors, pressure, humidity and temperature, which are linearly interpolated in time. This nudging is applied over the full 40 atmospheric layers used in this version. In the upper atmosphere, temperature, wind vectors and humidity are relaxed toward ERA5 following van de Berg & Medley, 2016. At the sea-surface interface, sea surface temperature and sea-ice extent from ERA5 are prescribed."

And for HCLIM as follows:
"HCLIM43 utilised ERA5 data at three-hour intervals to provide lateral boundary conditions and ocean surface forcing. This included inputs for temperature, wind components (zonal

and meridional), specific humidity, sea ice concentration, sea surface temperature, and surface pressure. Spectral nudging was implemented above 850 hPa for air temperature, divergence, and vorticity, employing a horizontal length scale of 800 km. The model incorporates the Simple Ice Scheme (SICE), a thermodynamic sea ice module that calculates the vertical temperature profile of sea ice (Batrak et al., 2018)."

Lines 183-184: Using daily data obscures the diurnal cycles that are prevalent in many parts of Antarctica during austral summer.

We chose to work with daily data to simplify handling of multiple RCM simulations. Analysis of diurnal cycles in the RCMs is beyond the scope of this study, which focuses on seasonal and interannual variability, and will be addressed in forthcoming studies targeting specific locations or case studies.

Line 194: Were the wind speeds adjusted to the same height above the surface? Typically, the models produce 10-m wind speeds while the AWS winds are typically at 3 m above the surface.

The observed 3m wind speed was indeed recalculated to 10m wind speed, this is now clarified in the methods as follows: "Besides, the observed wind speed measured at 3 m height is recalculated to 10 m assuming a logarithmic wind profile, enabling a direct comparison with the modelled 10 m wind fields."

Figure 1: Your analysis is heavily weighted to the Ross Island region and the Victoria Land coast.

Thank you for this comment. To accommodate this we have weighted the statistics shown in. Fig. 8 with the area that the AWS represents, to avoid that these statistics are weighted to the Ross island and Victoria Land coast. This is clarified in the Methodology section as follows: "Finally, to account for the uneven spatial distribution of AWSs, daily statistics are area-weighted according to Voronoi polygons around each station. Each station's contribution is proportional to the surface area it represents, preventing that the statistics are biased towards regions where the AWS network is denser, such as Victoria Land."

Figure 2: Please try a nonlinear scale for specific humidity and precipitation to provide more details over Antarctica, the focus of interest. The pressure plots are not useful as they are dominated by the effects of terrain height. The standard deviation plots for pressure are meaningful, however.

We have now used a non-linear scale for specific humidity and precipitation to improve the visibility of the plots. We have retained the MMM pressure plot here since the standard deviation plots for pressure give useful information on how topography differs between the RCMs which can impact the other variables.

Table 1: Give precipitation in mm/year, the unit used in the plots.

We have adapted this now.

Line 253: Specific humidity not relative humidity.

We have changed this.

Figure 3: Please use a nonlinear scale to provide more details over Antarctica. For context add panels for ERA5, like done for Figs. 5-7. You discuss sea ice in several locations, but nowhere do you display the sea ice extent. Please rectify.

We have applied a nonlinear scale to precipitation (Figure 4) and have added an ERA5 column to both Figure 3 and Figure 4. Sea ice extent from ERA5 is now indicated with a white contour in Figure 3.

Figure 4: Add ERA5 plots for context. Can the color bar be modified to provide more discrimination for larger values?

We have now applied the same non-linear scaling as in Figure 2 and have added a column for ERA5 for context.

Figure 5: For pressure some stations have unexpected large biases especially below the 1:1 line, around 10 hPa. These are probably station elevation errors. Please confirm.

We have now discussed these outliers for pressure as follows "A few stations show scatter consistently away from the 1:1 line for all RCMs and ERA5. These outliers are likely due to errors in the reported station heights, since the modelled pressure has been corrected for elevation differences with the AWSs."

Figure 7: This needs to be a full-page figure to see the details.

We have changed this.

Discussion of the biases in Figs. 6 and 7 on pages 16 and 17. Please adjust the language to make it clear you are discussing absolute biases.

We have changed this.

Figure 8: This is a nice figure. Please add the comparison for the ensemble mean. Correct line 423 to "(huss, g/kg)".

We have now added the ensemble mean to this figure. Not that after your earlier comment the statistics are now weighted to the area that the AWSs are representing to avoid bias to densely measured areas.

Figures 9 and 10: Nice figures and easily interpretable. Because you are emphasizing the ensemble mean this needs to be added.

An additional column for the multi-model mean is now included.

Figure 11: Please add corresponding figures to the Supplementary material stratifying by elevation.

Thank you for this comment. We add the plots.

Lines 503-504: "The lowest inter-quartile range is found in near surface air temperature and surface pressure in all seasons, particularly JJA." I don't think you can compare between variables because this depends on the units considered.

Thank you for this comment. We agree and remove this sentence.

Lines 507-508: "The RCMs shown here offer an improvement over ERA5 due to their higher resolution and more sophisticated physical parameterizations." The last statement cannot be claimed for HCLIM and is only somewhat true for RACMO that depends on parameterizations from ECMWF.

Thank you for this comment. We agree that this statement was too general. While RCMs benefit from higher spatial resolution relative to ERA5, the degree of added physical complexity depends on the specific model. We have therefore revised the text and now

emphasize the role of higher resolution and regional tailoring as the primary sources of added value relative to ERA5.

Lines 536-537: "No single RCM stands out as better or worse overall than any other". HCLIM is clearly deficient in relation to the other 3 models. Line 602 says that" HCLIM has not been used extensively in Antarctica before". HCLIM is not in the same class as the other 3 models that have been extensively applied in Antarctica, but with some effort it can achieve a similar level of performance.

Thank you for this comment. We agree that the original statement was too general. The text has been revised accordingly. While no single RCM consistently outperforms the others across all variables and seasons, HCLIM shows comparatively weaker performance in this evaluation.

Lines 594-597: "Some RCMs may be simulating more katabatic outflow (and hence colder temperatures) near this steep terrain (e.g. the MetUM), whereas the other RCMs may be simulating more compressive warming such as foehn winds, or adiabatic compression of katabatic flow." I did not understand the basis for this statement.

Thank you for this comment, we agree that it is not clear. This sentence was attempting to offer reasons for the differing biases between RCMs in this region of steep terrain. We have revised as follows: Several RCMs have opposing biases in this region of steep terrain, for example MAR and RACMO are warmer than ERA5 here, while HCLIM and the MetUM are colder. This implies differences in the way the various RCMs parameterise orographic flow and topographically induced temperature effects.